# The structure of a LAIR1-containing human antibody reveals a novel mechanism of antigen recognition

**Fu-Lien Hsieh, Matthew K Higgins\***

Department of Biochemistry, University of Oxford, Oxford, United Kingdom

**Abstract** Antibodies are critical components of the human adaptive immune system, providing versatile scaffolds to display diverse antigen-binding surfaces. Nevertheless, most antibodies have similar architectures, with the variable immunoglobulin domains of the heavy and light chain each providing three hypervariable loops, which are varied to generate diversity. The recent identification of a novel class of antibody in humans from malaria endemic regions of Africa was therefore surprising as one hypervariable loop contains the entire collagen-binding domain of human LAIR1. Here, we present the structure of the Fab fragment of such an antibody. We show that its antigen-binding site has adopted an architecture that positions LAIR1, while itself being occluded. This therefore represents a novel means of antigen recognition, in which the Fab fragment of an antibody acts as an adaptor, linking a human protein insert with antigen-binding potential to the constant antibody regions which mediate immune cell recruitment.

**\*For correspondence:** matthew. higgins@bioch.ox.ac.uk

**Competing interests:** The authors declare that no competing interests exist.

## Introduction

The antigen-binding sites of human antibodies commonly adopt similar structures, with the light and heavy chains each providing three hypervariable loops that combine to form a surface that is complementary to the epitope. While the sequences of these complementarity determining regions (CDRs) are highly variable, five of the six CDRs (L1, L2, L3, H1 and H2) can be classified into a number of relatively small sets, with similar lengths and architectures, and their structures are predictable from sequence (*Chothia et al., 1989*; *North et al., 2011*). In contrast, the third CDR loop of the heavy chain (CDR H3) is more structurally diverse, most likely due to its location close to the V(D)J recombination site (*Weitzner et al., 2015*). Human antibodies typically have CDR H3 lengths of 8–16 residues (*Zemlin et al., 2008*) while mouse antibodies have CDR H3 lengths of 5–26 residues (*Zemlin et al., 2003*).

However, recent years have seen the discovery of antibodies with major differences from the norm, in particular due to changes in the length of the third CDR of the heavy chain. A set of antibodies with broadly neutralizing potential against HIV is one such example. Here, the third CDR loop of the heavy chain is elongated, allowing it to reach through the glycan shield that surrounds the gp120 protein to bind an otherwise concealed epitope (*McLellan et al., 2011*; *Pancera et al., 2013*; *Pejchal et al., 2010*). Such antibodies are rare, making the induction of a broadly inhibitory response against HIV a major challenge (*Corti and Lanzavecchia, 2013*).

In a more extreme example, while the majority of bovine antibodies have CDR H3 loops of around 23 residues, around 10% contain a highly elongated third CDR loop of up to 69 residues, containing a small disulfide rich domain (*Saini et al., 1999*; *Wang et al., 2013*). These domains adopt a conserved $\beta$-sheet structure that displays variable loops and are each presented on an elongated, but rigid $\beta$-hairpin (*Stanfield et al., 2016*; *Wang et al., 2013*). While it is clear that the

**eLife digest** When bacteria, viruses or parasites invade the human body, the immune system responds by producing proteins called antibodies. Antibodies recognize and bind to molecules (known as antigens) on the surface of the invaders. This binding can either neutralize the invader directly or trigger signals that cause other parts of the immune system to destroy it.

Our blood contains a huge range of different antibody molecules that each bind to a different antigen. This is despite most human antibodies having the same basic shape and structure. Six loops, known as complementarity determining regions (CDRs), emerge from the surface of the antibody to form the surface that recognizes the antigen. However, variations in the structure of the loops alter this surface enough to allow different antibodies to recognize completely different molecules.

In 2016, a new class of antibodies was identified. Unlike previously identified antibodies, these molecules had an entire human protein, called LAIR1, inserted into one of their CDR loops. Members of this group of antibodies bind to a molecule, known as a RIFIN, that is found on the surface of human red blood cells that are infected with the parasite that causes malaria.

How do LAIR1-containing antibodies bind to their RIFIN targets? Hsieh and Higgins investigated this question by using a technique called X-ray crystallography to determine the structure of the antibody. This revealed that instead of binding directly to an antigen, all of the six CDR loops in the LAIR1-containing antibody bind to the LAIR1 insert. By doing so, LAIR1 is oriented in a manner that enables it to bind to the RIFIN molecule from the parasite.

This is the first known example of an antibody that recruits another protein to bind to an antigen rather than binding directly to the pathogen itself. A future challenge will be to see if other antibodies exist that use this mechanism and whether it can be employed to design new therapeutic antibodies.

additional domains play an important role in ligand binding, the remaining five CDR loops are also exposed and further studies are needed to see the contribution that they make (*Wang et al., 2013*).

A recent study identified a group of even more unusual human antibodies in malaria endemic regions of Africa (*Tan et al., 2016*). These antibodies were discovered through their capacity to agglutinate human erythrocytes infected with different strains of *Plasmodium falciparum*, and they bind to a subset of RIFIN proteins. These RIFINs are displayed by the parasite on infected erythrocyte surfaces and are of uncertain function (*Chan et al., 2014*; *Gardner et al., 1998*; *Kyes et al., 1999*). The antibodies show a remarkable adaptation with an intact 96 residue protein, LAIR1, inserted into the third CDR loop of the antibody heavy chain. Indeed, LAIR1 was shown to be essential for the antibody to interact with RIFINs (*Tan et al., 2016*). In this study, we reveal the structure of the Fab fragment of one of these antibodies, showing how LAIR1 is presented on the antibody surface and drawing conclusions about how this class of antibody can recognize its ligand.

## Results

We expressed the two chains that make up the Fab fragment of antibody MGD21 (*Tan et al., 2016*) in a secreted form from HEK293 cells. This antibody has a kappa light chain (VK1-8/JK5) and a heavy chain in which LAIR1 has been inserted into CDR H3. This fragment was purified and crystallised, allowing a dataset to be collected to 2.52 Å resolution. The structure was determined by molecular replacement using LAIR1 (*Brondijk et al., 2010*) and the Fab fragment of antibody OX117 (*Nettleship et al., 2008*) as search models. This identified two copies of the MGD21 Fab fragment in the asymmetric unit of the crystal. A model was built for residues 2–211 of the light chain and 1–351 (with 214–219 and 264–270 disordered) of the heavy chain (*Figure 1*, *Figure 1—figure supplement 1*, *Figure 1—figure supplement 2*, *Table 1*). The two Fab fragments adopt the same structure with a root mean square deviation of 0.26 Å (calculated over 475 Cα atoms) suggesting a highly ordered linkage between the variable domains of the antibody and the LAIR1 insert (*Figure 1—figure supplement 3*). The antibody sequence has three putative N-linked glycosylation sites, but of

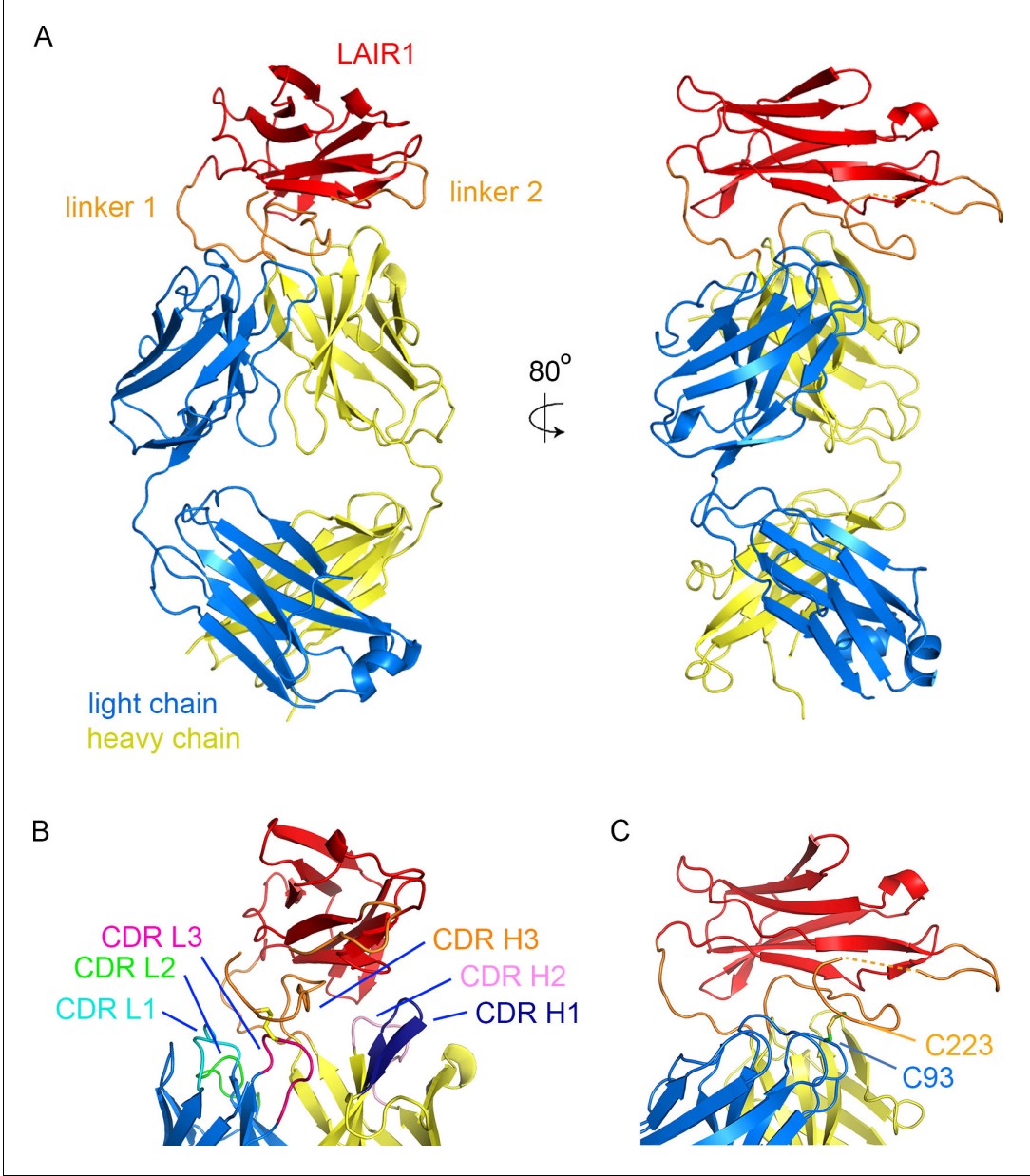

**Figure 1.** Structure of a LAIR1-containing antibody Fab fragment. (**A**) The structure of the Fab fragment. LAIR1 (red) is inserted into the third CDR loop of the heavy chain (yellow) through two extended linkers (orange). The light chain is blue. The dashed orange link represents protein disordered in the structure. (**B**) The organization of the CDRs. The three CDR loops of the light chain and remaining two CDR loops of the heavy chain directly contact the LAIR1 insert or the linkers. Each of the CDR loops and its corresponding label is a shown in a different colour. (**C**) A disulphide bond between C93 of the light chain and C223 of the heavy chain stabilizes the interface (cysteine residues are shown as sticks).

The following figure supplements are available for figure 1:

**Figure supplement 1.** Annotated sequence of antibody MGD21 and its alignment to germ line LAIR1.

**Figure supplement 2.** Electron density.

**Figure supplement 3.** Crystal packing and order.

**Table 1.** Data collection and refinement statistics. The structure was determined from a single crystal. Values in parentheses are for highest-resolution shell. $R_{free}$ was determined using 1968 reflections (4.8%) The structure is deposited with pdb code 5NST.

| | Fab-MGD21 |
|---|---|
| **Data collection** | |
| Space group | C121 |
| Cell dimensions | |
| *a, b, c* (Å) | 169.8, 86.5, 104.0 |
| $\alpha$ $\beta$ $\gamma$ (°) | 90.0, 126.7, 90.0 |
| Wavelength | 0.92819 |
| Resolution (Å) | 81.90–2.52 (2.56–2.52) |
| Total Observations | 131833 (5451) |
| Total Unique | 40946 (2031) |
| $R_{pim}$ (%) | 5.4 (67.8) |
| $R_{merge}$ (%) | 8.3 (88.5) |
| $R_{meas}$ (%) | 9.9 (112.1) |
| $CC_{1/2}$ | 0.992 (0.571) |
| $I/\sigma(I)$ | 7.4 (1.0) |
| Completeness (%) | 99.8 (98.3) |
| Multiplicity | 3.2 (2.7) |
| Wilson B factor | 55.216 |
| **Refinement** | |
| Number of reflections | 40946 |
| $R_{work}$ / $R_{free}$ | 21.9/26.7 |
| Number of residues | |
| Protein | 1076 |
| R.m.s deviations | |
| Bond lengths (Å) | 0.01 |
| Bond angles (°) | 1.25 |
| All Atom clash score | 5 |
| B factors | |
| All atoms | 71.53 |
| Solvent | 63.12 |
| Variable domains | 65.17 |
| Constant domains | 74.29 |
| LAIR1 insert | 73.70 |
| Linkers | 94.71 |
| Ramachandran plot | |
| Favored (%) | 95.2% |
| Allowed (%) | 4.8% |
| Disallowed (%) | 0.0% |

these (light chain N30; heavy chain N61 and N242) only N242 shows electron density corresponding to an Asn-linked N-acetyl glucosamine, in a position distant from the LAIR1 insert.

The structure shows LAIR1 emerging from the CDR3 loop of the heavy chain and lying across the antigen-binding surface of the variable domains of the Fab fragment (*Figure 1A*). The long axis of the LAIR1 insert is positioned with the β-strands aligned approximately perpendicular to the groove between the heavy and light chain CDRs and the insertion and linkers interact with, and occlude all five of the remaining CDR loops. The N- and C-termini of LAIR1 lie at opposite ends of its structure, necessitating long linkers between the sites from which CDR3 emerges from the antibody heavy chain and each terminus of the LAIR1 insert (*Figure 1A*). The N-terminal linker (linker 1) is 10 residues long and adopts a simple loop structure that joins the antibody variable domain to the N-terminus of the LAIR1 insert. The C-terminal linker (linker 2) is longer at 34 residues and is more complex in structure. It extends out from the C-terminus of the LAIR1 insert before zigzagging back towards the insertion site in the heavy chain variable domain. It is stabilized by hydrogen bonds to the LAIR1 insert and to the antibody heavy chain as well as by a disulphide bond to C93 of the antibody light chain. The linkers of the LAIR1-containing antibodies sequenced to date are variable both in length and content, involving different parts of the intronic regions of the LAIR1 gene, or intergenic sequences of chromosome 13 (*Tan et al., 2016*). The arrangement of these linkers, which radiate away from the remainder of the antibody, will in theory accommodate almost limitless variation in both length and sequence without disturbing the packing of LAIR1 against the variable domains of the antibody.

The five CDR loops lacking the LAIR1 insertion are representatives of previously identified canonical classes (*Figure 1—figure supplement 2*) (*Martin and Thornton, 1996*). However, a search using the Abcheck server (*Martin, 1996*) identified seven unusual residues within the antibody structure; C91, C93, D97 and I106 from the light chain and Y28, R34 and Q54 from the heavy chain, all within the CDR loops. In particular, C91, C93 and D97 all lie in CDR3 of the light chain, perhaps facilitating its interaction with linker 2. Indeed, the most unusual residue is C93, which is found in only 0.096% of light chains, and is the residue that forms a disulphide bond with linker 2 (*Figure 1B*). The heavy chain CDR H3 loop has a base that adopts the 'kinked' conformation (*Shirai et al., 1999*), with the loop rapidly spreading to the two termini of LAIR1.

In previous structures of antibodies with extended heavy chain CDR3 loops, the remaining five CDRs of the antibody are exposed, with the potential to engage in antigen binding (*McLellan et al., 2011*; *Stanfield et al., 2016*, *Wang et al., 2013*). One of the remarkable features of the LAIR1-containing antibody is therefore the occlusion of large parts of each of the remaining five CDRs, with these loops each interacting directly with the LAIR1 insert and/or linkers (*Figure 1B*, *Table 2*). The degree of occlusion of the CDRs by LAIR1 was determined by accessibility to a 1.4 Å probe in the presence and absence of LAIR1 and the linkers. Each of these five CDR loops was partly occluded by the presence of LAIR1 or the linkers (occluding 12.7% of the accessible surface area of CDR L1, 18.3% of CDR L2, 47.3% of CDR L3, 34.7% of CDR H1 and 16.0% of CDR H2). Indeed, both the first and second CDRs of the heavy chain directly contact the LAIR1 insert (*Table 2*). In addition, each of the three CDR loops of the light chain interacts with one of the two linkers, with interactions

**Table 2.** A list of interactions between the LAIR1 insert and linkers that occupies the heavy chain CDR3 loop and the other five CDR loops of the antibody.

| CDR loop | Residue | Group | LAIR1 region | Residue | Group | Interaction |
|---|---|---|---|---|---|---|
| Light chain CDR1 | Q27 | Side chain | Linker 2 | A222 | Main Chain | Hydrogen Bond |
| Light chain CDR2 | Y49 | Side chain | Linker 1 | L102 | Side chain | Hydrophobic Packing |
| Light chain CDR2 | N53 | Side chain | Linker 1 | S104 | Side chain | Hydrogen Bond |
| Light chain CDR3 | C93 | Side chain | Linker 2 | C223 | Side chain | Disulphide Bond |
| Light chain CDR3 | F94 | Main Chain | Linker 2 | E227 | Side Chain | Hydrogen Bond |
| Heavy chain CDR1 | N32 | Side chain | LAIR1 | R134 | Side chain | Hydrogen Bond |
| Heavy chain CDR2 | R57 | Side Chain | LAIR1 | P109 | Main Chain | Hydrogen Bond |

including a disulphide bond between C93 of the light chain and C223 of linker 2 (*Figure 1C*, *Table 2*). These interactions are replicated in both copies of the molecule in the asymmetric unit of the crystal.

The structure of MGD21 argues for a rigid association of the LAIR1 insert with the remainder of the antibody. Firstly, the structures of the two molecules of the antibody in the asymmetric unit of the crystal superimpose closely (*Figure 1—figure supplement 2*). It is unlikely that this is due solely to constraints from crystal packing as LAIR1 is anchored to the variable domains of the antibody through three fixed positions: the attachment sites of the two linkers, and the disulphide bond between light chain C93 and heavy chain C223 (*Figure 1C*). In addition, each of the five CDR loops not baring a LAIR1 insertion makes direct interactions with either LAIR1 or the linker, through contacts found in both copies of the antibody in the asymmetric unit of the crystal. This will stabilize a tight association between LAIR1 and the antibody variable domains. As these antibodies can include multiple different light chains, and very different linkers (*Tan et al., 2016*), these interaction will not be replicated precisely across the antibody family, but some variant of interaction between light chain CDRs and the linkers is likely.

A comparison of the LAIR1 insert with that of the chromosomal copy of LAIR1 (referred to below as germ line) (*Brondijk et al., 2010*) reveals that no global structural changes have taken place (root mean square deviation 0.43 Å for the 82 Cα residues) (*Figure 2A,F*). Indeed, the LAIR1 insertion in the MDG21 antibody differs in only 13 positions relative to the germ line sequence. Mapping these sites onto the structure reveals that they do not alter residues through which the LAIR1 insert interacts with the rest of the antibody (*Figure 2B*). Presumably, the stable interaction between LAIR1 and the antibody has therefore come instead from adaptations to the CDR loops. In contrast, polymorphisms are mostly located on the surface of LAIR1 distal to the rest of the antibody and have the potential to alter its interaction with its original ligand, collagen, and with the RIFIN proteins which are the target of these antibodies.

The normal function of LAIR1 is to interact with collagen (*Meyaard, 2008*). The structure of germ line LAIR1, together with NMR analysis and mutagenesis, allowed the mapping of residues critical for the collagen interaction onto a LAIR1 crystal structure (*Brondijk et al., 2010*). In particular, mutations in residues R59, E61 and R65 have a significant impact on collagen binding (*Brondijk et al., 2010*). These residues map onto the surface of LAIR1 (*Figure 2C*) that is most exposed in the context of the antibody (*Figure 2D*). Indeed, mapping of the polymorphisms found in the 27 LAIR1-containing antibodies sequenced to date shows that large parts of this surface are mutable (*Figure 2E*). The polymorphisms in LAIR1 include R149N, which is in the position equivalent to R65 in germ line LAIR1 and this change may impact collagen binding. A second polymorphism, found in 7/27 of the antibodies (including MGD21) alters the N-linked glycosyation site at residue 69 of LAIR1 (*Wollscheid et al., 2009*), which may alter collagen binding and/or increase RIFIN binding, but is not conserved across the antibody family. Indeed 11 of the 27 sequenced antibodies have mutations in at least one of the residues implicated in collagen binding, or other polymorphisms that reduce the interaction (*Tan et al., 2016*).

## Discussion

The LAIR1-containing antibodies are a remarkable variant of the standard immunoglobulin fold. While the majority of mammalian antibodies have predicable and short CDRs, the third CDR of the heavy chain can accommodate usual diversity (*Figure 3*) (*McLellan et al., 2011*; *Wang et al., 2013*; *Weitzner et al., 2015*). This is seen in the elongated CDR3 of the broadly neutralizing antibodies that interact with HIV surface proteins and in the insertion of a β-hairpin and disulphide-rich domain in a fraction of bovine antibodies. However, in both of these cases, only the heavy chain CDR3 is altered and the remaining CDR loops remain exposed for antigen binding. The LAIR1-containing antibodies are an exception to this, with the LAIR1-insert interacting with, and partly occluding, all five of the remaining CDR loops. In many ways, the structure resembles an antibody with CDR loops adapted for LAIR1 binding, into which LAIR1 has also been inserted.

This occlusion of large parts of the CDR loops by the LAIR1 insert has major consequences for its role in antigen recognition, as the majority of the antigen-binding surface will be contributed by LAIR1. Indeed, it has been shown that the LAIR1 insert alone can bind to infected erythrocytes, as can a LAIR1-containing antibody with the heavy and light chain regions exchanged (*Tan et al.,*

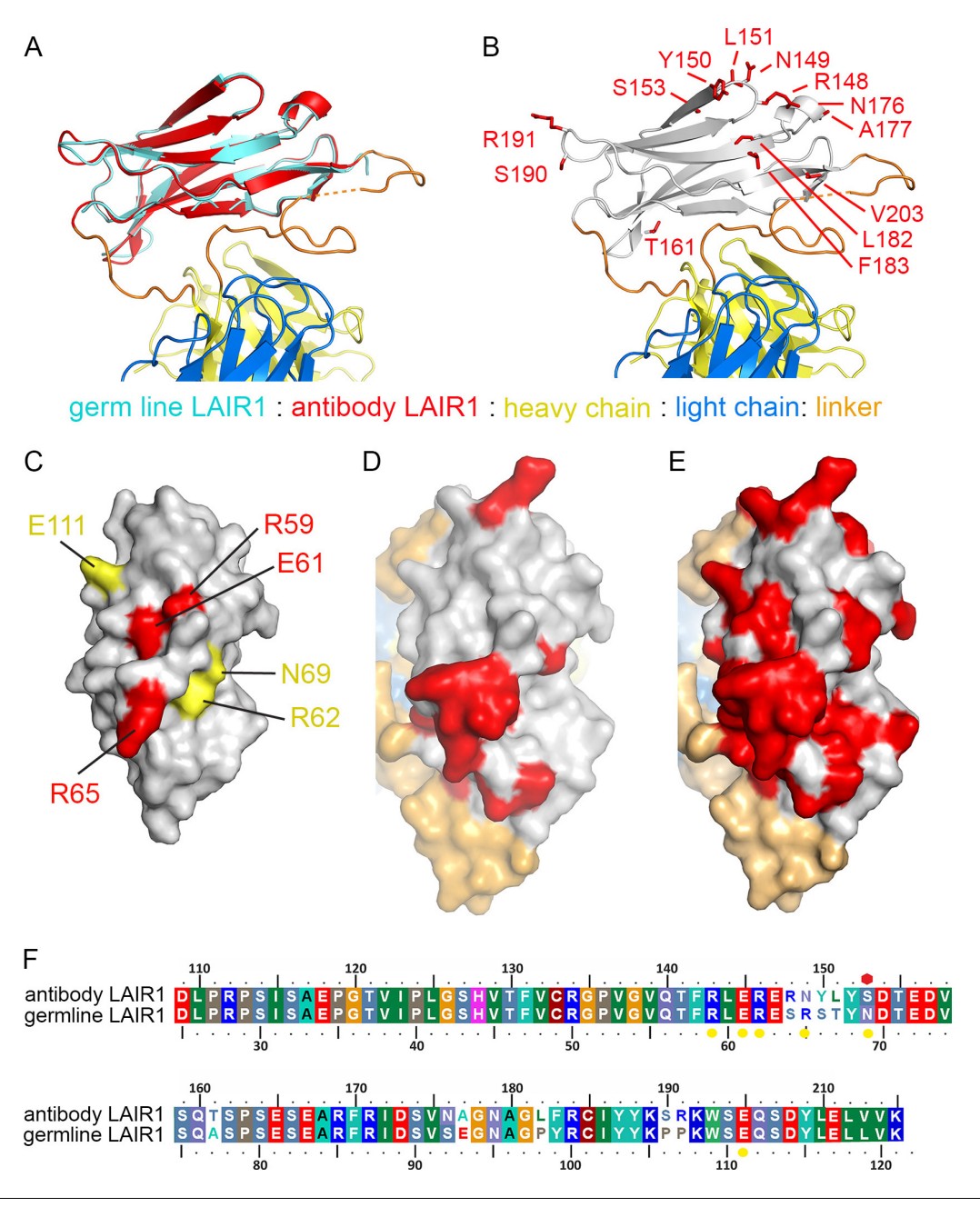

**Figure 2.** Structure and polymorphism in the LAIR1 insertion. (**A**) An alignment of germ line LAIR1 (cyan) with the antibody LAIR1 insertion (red). (**B**) The residues that differ between the LAIR1 insertion in antibody MGD21 and germ line LAIR1 are shown as red sticks. (**C**) A surface representation of the structure of LAIR1 (grey) with residues whose mutation has a major (red) or minor (yellow) effect on collagen binding highlighted (*Brondijk et al., 2010*). (**D**) A surface view of the LAIR1 insert in antibody MGD21 (grey) with residues that differ from germ line LAIR1 highlighted (red). (**E**) A surface view of the LAIR1 insert (grey) with residues that differ from germ line LAIR1 in all 27 antibodies tested to date (*Tan et al., 2016*) highlighted (red). (**F**) A sequence alignment of germ line LAIR1 and the LAIR1 insert in the MGD21 antibody. Yellow circles are sites residues shown to play a role in collagen binding while a red hexagon represents a potential N-linked glycosylation site mutated in the LAIR1 insert.

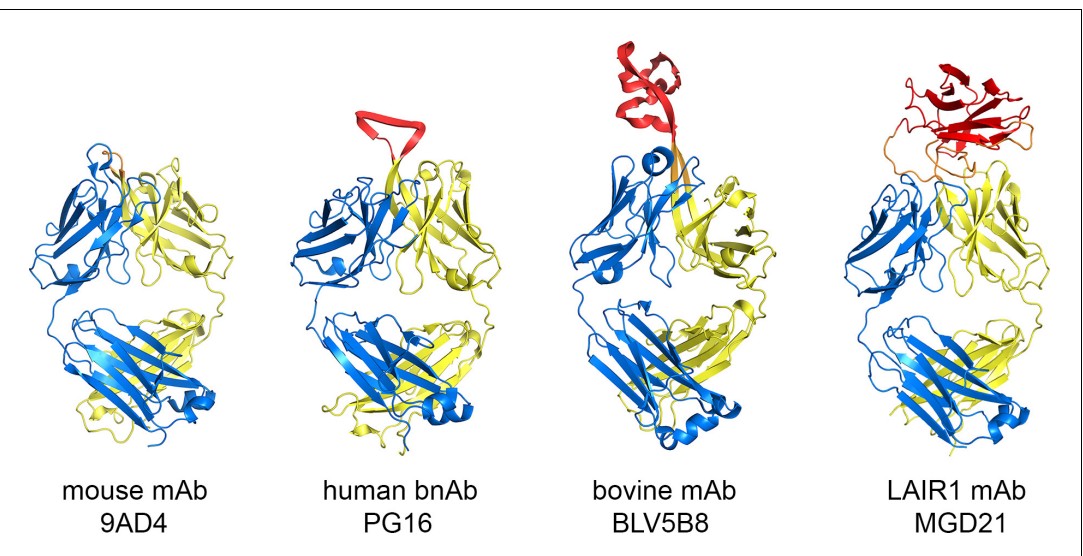

**Figure 3.** Comparison of the LAIR1-containing antibody with other unusual antibodies. The structure of the LAIR1-containing monoclonal antibody is compared with a classical mouse monoclonal antibody (9AD4; PDB code 4U0R), a human monoclonal antibody with broadly neutralizing potential against HIV (PG16; PDB code 4DQ0) and a bovine monoclonal antibody (BLV5B8; PDB code 4K3E). In each case, the light chain is blue and the two immunoglobulin domains of the heavy chain are yellow. Inserted domains are shown in red with linker regions in orange.

*2016*). Surprisingly an antibody in which the LAIR1 insert has been exchanged for the unaltered germ line LAIR1 did not bind to erythrocytes, although the folding of this chimera was not tested (*Tan et al., 2016*). In addition, the capacity of RIFINs to bind to unaltered LAIR1 alone has not yet been reported. Indeed, it seems most likely that LAIR1, or a highly related homologue, is the physiological ligand of the group of RIFINs that are recognized by these antibodies and that its insertion into the Fab fragment of an antibody allows it to be affinity matured to mobilise it for immune recognition and recruitment of immune cells. This remarkable LAIR1-containing antibody therefore uses the classical hypervariable loops for a novel function: to position an inserted auxiliary domain for antigen recognition. The classical Fab fragment therefore now acts as a link between a ligand for a pathogen surface receptor and the Fc region of the antibody with its immune recruitment capability. It will be fascinating to see if this is a paradigm that is repeated in other novel antibodies, as yet undiscovered.

## Materials and methods

### Construction, protein expression and purification

Two synthetic complementary DNA clones based on MGD21 (*Tan et al., 2016*) were obtained from GeneArt (ThermoFisher, UK). The heavy chain variable region was amplified using primers VH-F: 5'-GATGGGTTGCGTAGCTGAAGTGCAGCTGGTGGAAACAGGC-3' and VH-R: 5'-GGGTGTCG TTTTGGCGCTAGACACTGTCACGGTGGTGCC-3'. The light chain variable region was amplified using primers VL-F: 5'-GATGGGTTGCGTAGCTGCCATCAGAATGACCCAGAGCCCC-3' and VL-R: 5'-GTGCAGCATCAGCCCGCTTGATTTCCAGCCGGGTGCCC-3'. The resulting PCR products were cloned into pOPINVH (heavy chain variable region) and pOPINVL (light chain variable region) by In-Fusion cloning (Clontech, Mountain View, CA) (*Nettleship et al., 2008*). Therefore the variable domains from MGD21 were fused to the constant domains derived from the pOPINVH and pOPINVL vectors.

DNA constructs expressing heavy and light chains were mixed into a 1 to 1 ratio and co-transfected in HEK293T cells (ThermoFisher Scientific, UK) with polyethyleneimine in the presence of 5 µM kifunensine (*Aricescu et al., 2006*). After five days, conditioned media was dialysed against phosphate-buffered saline and purified by immobilised metal ion affinity chromatography using

TALON resin (Clontech, Mountain View, CA). The Fab heterodimer was further purified by size-exclusion chromatography using a Superdex 200 16/600 column (GE Healthcare Life Sciences) in 10 mM HEPES, pH 7.5 and 150 mM NaCl.

## Crystallisation, data collection and structure determination

Concentrated protein (10 mg/ml) was incubated with *Flavobacterium meningosepticum* endoglycosidase-F1 for in situ deglycosylation (*Hsieh et al., 2016*). The protein samples were then subjected to sitting drop vapour diffusion crystallisation trials in SwisSci 96-well plates by mixing 100 nl protein with 100 nl reservoir solution. The protein crystals were obtained in 20% (w/v) PEG4000, 0.1 M sodium citrate, pH 4.5 at 18°C. Crystals were transferred into mother liquor containing 25% (w/v) glycerol and were then cryo-cooled in liquid nitrogen for storage and data collection. Data were collected on beamline I04-1 at the Diamond Light Source and were indexed and scaled using XDS (*Kabsch, 2010*). Phaser (*McCoy et al., 2007*) was used to determine a molecular replacement model, using the known structures of LAIR1 (pdb: 3KGR (*Brondijk et al., 2010*)) and a human monoclonal antibody Fab fragment similar to MGD21 (pdb: 3DIF, (*Nettleship et al., 2008*)) separated into two files containing the variable and the constant regions, as search models. This identified two copies of the LAIR1-containing Fab fragment in the asymmetric unit of the antibody. Refinement and rebuilding was completed using Buster (*Blanc et al., 2004*) and Coot (*Emsley et al., 2010*) respectively. To determine the effect of the LAIR1 insert on the accessible surface area of the CDR loops, we used AREAIMOL from the CCP4 suite (*Winn et al., 2011*) to determine the accessible surface area of each CDR loop both in the presence and absence of LAIR1 and the linkers.

## Acknowledgements

We thank Dr. Ray Owens at the Oxford Protein Production Facility for the provision of vectors pOPINVH and pOPINVL and the beamline scientists at Diamond Light Source for assistance with data collection.

## Additional information

### Funding

| Funder | Grant reference number | Author |
|---|---|---|
| Wellcome | 101020/Z/13/Z | Fu-Lien Hsieh<br>Matthew K Higgins |
| Taiwan Bio-Development Foundation | | Fu-Lien Hsieh |

The funders had no role in study design, data collection and interpretation, or the decision to submit the work for publication.

### Author contributions

F-LH, Conceptualization, Formal analysis, Investigation, Writing—original draft; MKH, Conceptualization, Formal analysis, Supervision, Funding acquisition, Investigation, Methodology, Writing—original draft, Project administration

### Author ORCIDs

Matthew K Higgins, http://orcid.org/0000-0002-2870-1955

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
