## [Decision Letter]

Thank you for submitting your article "The structure of a LAIR1-containing human antibody reveals a novel mechanism of antigen recognition" for consideration by *eLife*. Your article has been favorably evaluated by Michel Nussenzweig (Senior Editor) and three reviewers, one of whom is a member of our Board of Reviewing Editors. The following individual involved in review of your submission has agreed to reveal her identity: Robyn Stanfield (Reviewer #2).

The reviewers have discussed the reviews with one another and the Reviewing Editor has drafted this decision to help you prepare a revised submission.

Summary:

This paper reports a very interesting structure, that of a LAIR1-containing Fab from a human antibody. This is possibly one of the most interesting Fab structures ever reported, and the paper could become a real classic if the authors will include a more detailed analysis of the structure and comparison of its features with those from 'normal' Fabs. We have the following suggestions and concerns:

Essential revisions:

1) The most interesting part of the new structure is that the inserted LAIR1 domain occludes the antibody's combining site, which is relevant if the LAIR1 domain is rigidly anchored in such a way to truly preclude access to the CDRs. However, the manuscript makes the point that the CDR H3 linkers are long and flexible, so perhaps the occlusion of the combining is not obligate, but represents the arrangement of the LAIR1 domain in this crystal packing. This should be addressed by an analysis of the different crystal packing environments for the two Fabs in the asymmetric unit of the crystals and a discussion of whether the LAIR1 addition shows the same interactions with the Fab in both copies. A deeper discussion of whether the LAIR1 domain is rigidly docked to the Fab or not, including consideration of crystal contacts limiting domain movement, should be added to a revised manuscript.

2) Related to point 1, the authors can also address whether the LAIR1 domain is rigidly docked to the Fab by discussing whether there are any unusual residues in the Fab framework regions that might be important for interactions with the LAIR domain. Andrew Martin has a website that will identify unusual amino acids in Fabs.

3) Also related to point 1, the authors should show surface areas accessible to a 1.4 Å probe on the CDRs for the structures shown in Figure 3. This would be a nice way to demonstrate that the CDRs are inaccessible in the LAIR1 mAb but not in the other Fab structures. Also, what is the buried surface area between Fab and LAIR1 insert? It would be informative to show these numbers as a function of CDRs (i.e., how much surface is buried at each CDR, and how much at the framework, and on the other side how much of LAIR 1/insert are buried). It was hard to tell from figures and text if the LAIR1 domain is really binding to the CDRs or is just suspended over them? Please explain in more detail.

4) Do the CDR L1, L2, L3, H1 and H2 structures belong to the expected canonical classes?

5) Is the light chain a lambda or kappa chain?

6) Is the H3 base conformation kinked or extended or something else? The Asn at H242 that may be glycosylated can play a role in the conformation of the H3 base, so might see if potential glycosylation there has altered the expected structure.

7) Do you see any residual Nag density? Fabs can by glycosylated, but I don't recall ever seeing a Fab with 3 N-linked glycosylation sites, so could you comment on whether the full-length glycans would likely interact with the LAIR1, or with the RIFIN antigen? And would the germline LAIR1 glycan (at H153) have clashed with the Fab, or would it have been pointing outward toward antigen?

8) In Figure 1—figure supplement 1, it would be helpful to show under the sequences the alignment of the putative V, D, and J genes- to show for example in the H3 region which residues in the linker were likely derived from LAIR1 gene, and which residues derived from antibody gene segments. Also, in addition to the sequential numbers, please show the Kabat numbering for light chain and for the heavy chain (except for the LAIR1 insert which will be impossible to number by the Kabat system).

9) Refinement: The *R_free_* is a bit high and the Ramachandran statistics are not great for a 2.5 Å structure. Additional refinement cycles and/or rebuilding should be done. Also, Table 1 is too brief. Please include the PDB code, # of observations/unique reflections collected, *Rmerge, R_meas_*, mean I/σ, # of reflections in the *R_free_* test set, the B values (Wilson B, and overall B's for solvent, variable, constant, and LAIR1 domains), and the all-atom clashscore. Please also include at least one figure in the supplemental material showing some electron density.

10) Figures:

Figure 1. CDRs should be highlighted in colors so they can be distinguished. Linkers to LAIR1 should also be highlighted.

Figure 1. Disulfide bond should be highlighted with the cysteines as sticks.

Figure 2 legend. Proper terminology is "potential N-linked glycosylation site" rather than "putative glycosylation site." Position of the Asn (serine in antibody LAIR1) should be highlighted on the structure.

Figure 2. Residues shown as sticks should be labeled on the figure.

Figure 3. CDRs should be highlighted on all structures.

---

## [Author Response]

*Essential revisions:*

*1) The most interesting part of the new structure is that the inserted LAIR1 domain occludes the antibody's combining site, which is relevant if the LAIR1 domain is rigidly anchored in such a way to truly preclude access to the CDRs. However, the manuscript makes the point that the CDR H3 linkers are long and flexible, so perhaps the occlusion of the combining is not obligate, but represents the arrangement of the LAIR1 domain in this crystal packing. This should be addressed by an analysis of the different crystal packing environments for the two Fabs in the asymmetric unit of the crystals and a discussion of whether the LAIR1 addition shows the same interactions with the Fab in both copies. A deeper discussion of whether the LAIR1 domain is rigidly docked to the Fab or not, including consideration of crystal contacts limiting domain movement, should be added to a revised manuscript.*

The reviewers start by asking if the arrangement of LAIR1 relative to the rest of the Fab fragment is obligate. Our structure suggests that it is, and we have included a deeper discussion of the evidence in the manuscript. Firstly, when we align the two copies of the antibody structure, they agree extremely well, with an RMSD of 0.26Å, as reported in the Results section. We now include a figure (Figure 1—figure supplement 3) to show this alignment. This shared structure is very unlikely to be due to crystal packing due to the interactions that LAIR1 and the linker make with the remaining CDR loops. In addition, the presence of a disulphide bond between light chain C93 and heavy chain C223 is seen in both copies in the asymmetric unit (labelled in Figure 1). This provides a third anchor point between LAIR1 and the rest of the antibody (together with the junctions with the linkers) and will stabilise their relative organisation. Finally, the interactions between each of the antibody CDRs and the LAIR1 insert and linker (as outlined in Table 2) are replicated in both copies of the asymmetric unit. This evidence all points to a rigid association between the antibody and the LAIR1 insertion. We have collected this argument together into a single place in the Results section to make it clearer.

*2) Related to point 1, the authors can also address whether the LAIR1 domain is rigidly docked to the Fab by discussing whether there are any unusual residues in the Fab framework regions that might be important for interactions with the LAIR domain. Andrew Martin has a website that will identify unusual amino acids in Fabs.*

Use of Andrew Martin’s server identified seven amino acids which are usual in this antibody. Four of these occupy the third CDR of the light chain. The most unusual of the residues is light chain C93, which we had previously discussed as forming a disulphide bond with the LAIR1 insert, and is only found in 0.096% of antibody chains. We have added three sentences to the Results section to describe this analysis and have cited the paper related to this server in the paper.

*3) Also related to point 1, the authors should show surface areas accessible to a 1.4 Å probe on the CDRs for the structures shown in Figure 3. This would be a nice way to demonstrate that the CDRs are inaccessible in the LAIR1 mAb but not in the other Fab structures. Also, what is the buried surface area between Fab and LAIR1 insert? It would be informative to show these numbers as a function of CDRs (i.e., how much surface is buried at each CDR, and how much at the framework, and on the other side how much of LAIR 1/insert are buried). It was hard to tell from figures and text if the LAIR1 domain is really binding to the CDRs or is just suspended over them? Please explain in more detail.*

As suggested, we determined the percentage of each of the CDR loops that is occluded by LAIR1 and the linkers. To achieve this, we used AREAIMOL from the CCP4 suite to calculate the accessible surface area of each CDR loop to a 1.4Å sphere, both in the intact antibody and an antibody in which we removed CDR H3. This confirmed that each of the five remaining CDR loops was partly occluded by the presence of LAIR1. We have provided this information in the Results section, together with the percentage occlusion, and have also described the procedure used in the Materials and methods.

*4) Do the CDR L1, L2, L3, H1 and H2 structures belong to the expected canonical classes?*

The five CDR loops which do not bare the extension are all members of the canonical classes of CDRs (CDR L1 class 2/11AA; CDR L2 class 1/7AA; CDR L3 class 1/9AA; CDR H1 class 2/11AA; CDR H2 class 1/16AA). We have provided this information in Figure 1—figure supplement 1, in the boxes that indicate the locations of the CDR loops.

*5) Is the light chain a lambda or kappa chain?*

The light chain of MGD21 is kappa (VK1-8/JK5). We have provided this information in the first paragraph of the Results section.

*6) Is the H3 base conformation kinked or extended or something else? The Asn at H242 that may be glycosylated can play a role in the conformation of the H3 base, so might see if potential glycosylation there has altered the expected structure.*

We have confirmed that the base of the H3 loop is of the kinked conformation, as defined by the rules of Shirai et al. We have mentioned this in the text.

*7) Do you see any residual Nag density? Fabs can by glycosylated, but I don't recall ever seeing a Fab with 3 N-linked glycosylation sites, so could you comment on whether the full-length glycans would likely interact with the LAIR1, or with the RIFIN antigen? And would the germline LAIR1 glycan (at H153) have clashed with the Fab, or would it have been pointing outward toward antigen?*

The reviewers ask about the glycosylation state of the antibody chains. Two of the putative N-linked glycosylation sites (light chain N30 and heavy chain N61) show no evidence of glycosylation in the electron density, while a single GlcNAc is observed on N242 of the heavy chain in each of the copies in the asymmetric unit. We have added a sentence to the first paragraph of the Results section to provide this information and have now shown the sugars in stick representation on the structure in Figure 1.

The reviewers also ask about the location of residue 153 of the heavy chain as the equivalent site in germ line LAIR1 is modified by N-linked glycosylation (residue N69) and this Asn has been mutated to Ser in the form of LAIR1 inserted into the antibody. This residue points away from the rest of the Fab and had already been highlighted in Figure 2 as a residue whose mutation has a minor effect on collagen binding by LAIR1. It is possible that this glycosyation site has been lost to increase RIFIN binding, but in ~75% of the LAIR-1 containing antibody sequences reported to date the glycosylation site is intact. We have now discussed this in the last paragraph of the Results section.

*8) In Figure 1—figure supplement 1, it would be helpful to show under the sequences the alignment of the putative V, D, and J genes- to show for example in the H3 region which residues in the linker were likely derived from LAIR1 gene, and which residues derived from antibody gene segments. Also, in addition to the sequential numbers, please show the Kabat numbering for light chain and for the heavy chain (except for the LAIR1 insert which will be impossible to number by the Kabat system).*

We have updated Figure 1—figure supplement 1 to label the regions of the heavy chain that derive from the V and J genes, and from the LAIR1 intron. We have also labelled the sequences of both light and heavy chains with Kabat numbers.

*9) Refinement: The R_free_ is a bit high and the Ramachandran statistics are not great for a 2.5 Å structure. Additional refinement cycles and/or rebuilding should be done. Also, Table 1 is too brief. Please include the PDB code, # of observations/unique reflections collected, Rmerge, R_meas_, mean I/σ, # of reflections in the R_free_ test set, the B values (Wilson B, and overall B's for solvent, variable, constant, and LAIR1 domains), and the all-atom clashscore. Please also include at least one figure in the supplemental material showing some electron density.*

We have completed the refinement of the structure with significant improvements in the Ramachandran statistics. We have deposited it in the PDB with code 5NST and a final validation report is included in this submission. We have also updated Table 1 to include the additional statistics requested by the reviewers. For note, we chose to use the CC_1/2_>0.5 cut off for data processing after observing increased detail in the electron density maps by including the extra data. We have also included a new figure (Figure 1—figure supplement 2) to show electron density for the whole antibody as well as two important regions of the molecule where CDR loops contact and stabilise the LAIR1 insertion.

*10) Figures:*

Figure 1. CDRs should be highlighted in colors so they can be distinguished. Linkers to LAIR1 should also be highlighted.

Figure 1. Disulfide bond should be highlighted with the cysteines as sticks.

Figure 2 legend. Proper terminology is "potential N-linked glycosylation site" rather than "putative glycosylation site." Position of the Asn (serine in antibody LAIR1) should be highlighted on the structure.

Figure 2. Residues shown as sticks should be labeled on the figure.

*Figure 3. CDRs should be highlighted on all structures.*

We have made various changes to the figures to improve their clarity and to provide the information requested by the reviewers.

We prefer to keep Figure 1 unchanged for clarity but have ensured the linkers are highlighted in orange and are labelled.

We have redrawn Figure 1. The disulphide bond is shown with the cysteine residues as sticks (as it is in Figure 1) and we have given each of the CDR loops a different colour, which matches the colour of their label.

We have labelled each of the residues shown as sticks in Figure 2 and have altered the legend with regard to the description of the glycosylation site.

We prefer to keep Figure 3 unchanged, rather than labelling the individual CDRs, as this keeps the figure ‘cleaner’ and focuses the attention of the reader more onto the key region of the CDR H3 loop that we are aiming to highlight here.